# Proteomic Profiling of *Paracoccidioides brasiliensis* in Response to Phenacylideneoxindol Derivative: Unveiling Molecular Targets and Pathways

**DOI:** 10.3390/jof9080854

**Published:** 2023-08-16

**Authors:** Lívia do Carmo Silva, Olivia Basso Rocha, Igor Godinho Portis, Thaynara Gonzaga Santos, Kleber Santiago Freitas e Silva, Raimundo Francisco dos Santos Filho, Silvio Cunha, Antônio Alonso, Célia Maria de Almeida Soares, Maristela Pereira

**Affiliations:** 1Institute of Biological Sciences, Universidade Federal de Goiás, Goiânia 74690-900, Brazil; oliviabassorocha@gmail.com (O.B.R.); igorportis@gmail.com (I.G.P.); thaynara075@hotmail.com (T.G.S.); smallbinho@hotmail.com (K.S.F.e.S.); cmasoares@gmail.com (C.M.d.A.S.); maristelaufg@gmail.com (M.P.); 2Institute of Chemistry, Universidade Federal da Bahia, Salvador 40170-970, Brazil; aimundo.santos@ifto.edu.br (R.F.d.S.F.); silviodc@ufba.br (S.C.); 3Institute of Physics, Universidade Federal de Goiás, Goiânia 74690-900, Brazil; alonso@ufg.br

**Keywords:** paracoccidioidomycosis, antifungal, nitrogen heterocycles, phenacylideneoxindol proteomic

## Abstract

Background: The treatment of paracoccidioidomycosis (PCM) is a challenge, and the discovery of new antifungal compounds is crucial. The phenacylideneoxindoles exhibited promising antifungal activity against *Paracoccidioides* spp., but their mode of action remains unknown. Methods: Through proteomic analysis, we investigated the effects of (E)-3-(2-oxo-2-phenylethylidene)indolin-2-one on *P. brasiliensis*. In addition, we investigated the metabolic alterations of *P. brasiliensis* in response to the compound. Furthermore, the effects of the compound on the membrane, ethanol production, and reactive oxygen species (ROS) production were verified. Results: We identified differentially regulated proteins that revealed significant metabolic reorganization, including an increase in ethanol production, suggesting the activation of alcoholic fermentation and alterations in the rigidity of fungal cell membrane with an increase of the ergosterol content and formation of ROS. Conclusions: These findings enhance our understanding of the mode of action and response of *P. brasiliensis* to the investigated promising antifungal compound, emphasizing its potential as a candidate for the treatment of PCM.

## 1. Introduction

*Paracoccidioides* is a genus of thermally dimorphic fungi that causes Paracoccidioidomycosis (PCM), an endemic and systemic mycosis in Latin America. PCM represents a public health problem with a significant impact on the quality of life of affected individuals. A higher incidence of PCM is associated with rural and agricultural areas or occupational exposure to environmental sources, such as soil and vegetation. Additionally, specific demographic groups are more commonly affected by PCM, such as adult males between 30 and 59 years old [1].

The PCM treatment aims to eradicate the *Paracoccidioides* fungus from the body, as well as to reduce symptoms, prevent recurrence, and improve the quality of life of patients. Although *Paracoccidioides* is sensitive to most antifungals, only a few are currently clinically employed [2]. Azole derivatives, such as itraconazole, have been widely used as first-choice therapy for mild to moderate forms of the disease. These drugs act by inhibiting the synthesis of ergosterol, an essential component of the fungal cell membrane, leading to the death of the fungus [3,4]. The duration of the treatment varies from months to years, depending on the clinical form of PCM and the individual patient’s response. In the treatment of disseminated PCM, which affects vital organs, intravenous administration of amphotericin B, a broad-spectrum polyene antifungal, is employed. This antifungal drug acts by binding to the ergosterol in the fungal cell membrane, causing its disruption and subsequent cell death [5,6]. However, it is important to note that amphotericin B can lead to severe side effects [7] and is typically reserved for cases in which other antifungal agents have demonstrated ineffectiveness or cannot be utilized due to contraindications. The combination of sulfamethoxazole and trimethoprim, which inhibit multiple steps of folic acid synthesis, has been recommended for less severe cases of the disease, mainly for the chronic and non-disseminated forms [1].

The treatment of PCM has some limitations that affect the therapeutic efficacy and quality of life of patients. The prolonged duration of treatment, ranging from months to years depending on the clinical form of the disease and the individual patient`s response, can impact adherence, resulting in interruptions in the therapeutic regimen, thereby compromising the treatment’s effectiveness. Adverse effects of these antifungals, such as kidney and liver damage, have been reported [8], requiring monitoring during the treatment period. Antifungal resistance is another concern. Although resistance is still relatively uncommon in clinical settings, the possibility of developing resistance was reported by an in vitro approach [9].

Considering the need to identify new compounds with antifungal activity against PCM, we previously evaluated the activity of five phenacylideneoxindoles compounds against several *Paracoccidioides* species [10]. The phenacylideneoxyndoles are nitrogenous heterocyclic compounds with a chemical structure that includes an oxindole ring with a phenacylidene group attached to it. These compounds possess bioactive properties and have been studied due to their pharmacological potential, including antifungal [11,12], antioxidant [13], antiparasitic [14], and anticancer activities [15]. One of the main reasons for studies is the potential for modifying its structure by incorporating functional groups, which enables the optimization of pharmacological properties, including affinity for specific targets, selective biological activity, and reduced toxicity. Additionally, the presence of nitrogen in the heterocyclic structure can facilitate important chemical and biological interactions with proteins and enzymes in the body, allowing for the modulation of metabolic pathways and physiological processes [16]. 

In our study with phenacylideneoxindoles [10], the compound (E)-3-(2-oxo-2-phenylethylidene)indolin-2-one (OPI) exhibited promising antifungal activity, displaying fungicidal effects and synergistic activity with itraconazole. In addition to conducting in vitro experiments, we also performed in silico predictions of potential molecular targets for OPI. However, we recognize the need to further advance research on the molecular alterations underlying its antifungal effect. Therefore, through proteomic analyses, we aimed to elucidate the impact of OPI on metabolic pathways and cellular processes. This article will provide insights into the differential expression of proteins in *Paracoccidioides* after exposure to OPI and demonstrate its effects on the cell membrane, ethanol levels, and production of reactive oxygen species (ROS).

## 2. Materials and Methods

### 2.1. Microorganism and Culture Conditions

*P. brasiliensis* (Pb18) was cultured in liquid Fava-Netto medium (0.3% protease peptone, 1% peptone, 0.5% (*w/v*) meat extract, 0.5% (*w/v*) yeast extract, 1% brain heart infusion, 4% glucose, 0.5% NaCl, 5 μg/mL gentamycin), pH 7.2, at 37 °C for 48 h with shaking. Subsequently, the cells were centrifuged at 5000× *g*, washed with phosphate-buffered saline–PBS (0.09% Na_2_HPO_4_, 0.02% KH_2_PO_4_, 0.8% NaCl, 0.02% KCl, pH 7.2), and then cultured again in Roswell Park Memorial Institute medium (RPMI 1640) for 16 h at 37 °C.

### 2.2. Phenacylideneoxindol Synthesis

OPI was synthesized according to the method described by Silva et al. [10]. A solution containing the mixture of 4.1 g (4 mmol) of acetophenone and 3.5 mL (34 mmol) of diethyl amine in 10 mL of ethyl alcohol was added dropwise to a solution of 5.0 g (34 mmol) of isatin in 30 mL of ethyl alcohol, under magnetic stirring at room temperature for 24 h. After this period, the resulting solid was filtered in a Büchner funnel, washed with a cold mixture (1:1) of water/ethyl alcohol, and dried at room temperature, yielding the intermediate 3-hydroxy-3-phenacyloxindole. A solution of 5.5 g (22 mmol) of 3-phenacyloxindole in 30 mL of 5% ethanolic HCl was subjected to magnetic stirring for 24 h. Following this step, the solid (E)-3-(2-oxo-2-phenylethylidene)indolin-2-one was filtered using a Büchner funnel, washed with cold ethyl alcohol, and dried at room temperature. The overall yield was 48%.

### 2.3. Cell Viability

For the cell viability assay, 1 × 10^5^ yeast cells/mL was incubated in the absence and presence of 7.62 µM OPI for 3, 6, 9, 12, 24, 48, and 72 h. At each time point, an aliquot of 1 mL was collected, and cell viability was measured using flow cytometry. The cells were incubated with propidium iodide (1 µg/mL) for 20 min in the dark at room temperature and then analyzed using a C6 Accuri flow cytometer (Accuri Cytometers, Ann Arbor, MI, USA). A total of 10,000 events per sample was acquired.

### 2.4. Proteomic Analysis

After fungal cultivation in Fava Netto’s medium, the cells were transferred to RPMI 1640 medium containing 7.62 µM of OPI and incubated for 9 h at 37 °C under shaking. For the control sample, cells were also cultured in RPMI 1640 medium without OPI and subjected to the same incubation conditions. The cells were then centrifuged, and the pellet was resuspended in ammonium bicarbonate buffer (57 mM, pH 8.8) with 20 µL of protease inhibitor (GE Healthcare, Uppsala, Sweden) and submitted to mechanical cell lysis using glass beads as described by Rocha et al. 2022 [17]. The cell lysate was centrifuged at 5000× *g* for 15 min at 4 °C to obtain the supernatant containing the protein extract. The integrity of the protein extract was verified by SDS-PAGE, and quantification was performed using the Bradford reagent (Sigma-Aldrich, St. Louis, MO, USA). We performed biological triplicates in proteomic analysis. The fungus *P. brasiliensis* was incubated in the presence and absence of the compound on different days. After obtaining the triplicates, the samples were pooled and subsequently digested. The resulting pool was analyzed in triplicate in the mass spectrometry analyses. 

The proteins were digested according to Rocha et al. 2022 [17] using RapiGEST^TM^ 0.2% (Waters Corporation, Milford, MA, USA). The nanoscale LC separation of tryptic peptides was performed using a nanoACQUITY^TM^ system (Waters Corporation, Milford, MA, USA). The peptides were separated into five fractions: 10.8%, 14%, 16.7%, 20.4%, and 65% of acetonitrile/0.1% (*v/v*) formic acid, with a flow rate of 2000 µL/min. The source was operated in positive ionization mode nano-ESI (+). To perform external calibration, the masses were corrected based on Glu-fibrinopeptide B (GFP; Sigma-Aldrich, St. Louis, MO, USA) of molecular mass 785.8486. We used a GFP solution in 50% (*v/v*) methanol and 0.1% (*v/v*) formic acid at a final concentration of 200 fmol/µL delivered by the reference sprayer of the NanoLockSpray source of the mass spectrometer. Mass spectrometry was analyzed on a Synapt G1 MS^TM^ (Waters, Milford, MA, USA) equipped with a NanoElectronSpray source and two mass analyzers: a quadrupole and a time-of-flight (TOF) operating in V-mode. Data were obtained using the instrument in the MS^E^ mode. Samples were analyzed from three replicates.

MS raw data were processed using the ProteinLynx Global Server version 2.4 (Waters Corporation, Milford, MA, USA). The data were subjected to automatic background subtraction, deisotoping, and charge state deconvolution. After processing, each ion comprised an exact mass-retention time that contained the retention time, intensity-weighted average charge, inferred molecular weight based on charge, and *m/z*. Then, the processed spectra were searched against *Pb*18 protein sequences (Broad Institute; http://www.broadinstitute.org/annotation/genome/Paracoccidioides_brasiliensis/Multiome.html, accessed on 15 April 2020) together with reverse sequences. The proteins were classified as induced or repressed after exposure to OPI, considering a fold change value of 1.3. Proteins were categorized into metabolic functions using the fungiDB and Functional Catalogue (FunCat). The submission of the proteomics results was assigned the identifier PASS05839 in the PeptideAtlas repository.

### 2.5. Ethanol Dosage

The ethanol production was measured and compared in cells, exposed or not to OPI, after 12 h of incubation and using the colorimetric detection kit (Sigma-Aldrich, St. Louis, MO, USA) following the manufacturer’s instructions. For the analysis, 1 mL of sample was collected and performed in biological triplicate. The statistical difference was evaluated using Student’s *t*-test and considered significant when *p* ≤ 0.05.

### 2.6. Spin Labeling and Electron Paramagnetic Resonance (EPR) Spectroscopy

After incubation of 9 h in the presence or absence of OPI, 2 mL of each sample were centrifuged at 25,000× *g* and 4 °C for 10 min. The supernatant was then removed, and the cells were suspended in 2 mL of PBS 1× and centrifuged again. The pellet volume of each sample was adjusted to 50 µL using PBS 1×. Next, the lipid spin label 5-doxy stearic acid (5-DSA) (Sigma-Aldrich, St. Louis, MO, USA) was incorporated in the fungal membranes following a previously described method [18]. Briefly, a spin-label film was prepared on the bottom of a glass tube using a 1 μL aliquot of an ethanolic solution containing 5-DSA at a 4 mg/mL concentration. After solvent evaporation, 50 μL samples containing ~3 × 10^6^ fungal cells were added to the spin-label film and gently agitated. For the EPR measurements, the spin-labeled sample was transferred to a 1-mm-i.d. capillary tube, which was sealed with a flame and then centrifuged at 25,000× *g* and 4 °C. The EPR spectra were recorded using an EPR EMX-Plus spectrometer (Bruker, Rheinstetten, Baden-Württemberg, Germany) and using the following instrumental settings: microwave power, 2 mW; modulation frequency, 100 kHz; modulation amplitude, 1.0 G; magnetic field scan, 100 G; sweep time, 168 s; and sample temperature, 25 °C.

### 2.7. Ergosterol Dosage

The ergosterol quantification was performed following the protocol described by Oliveira et al. [19]. Five grams of cells were obtained after 24 and 48 h of exposure, with or without OPI, and added to 5 mL of 25% alcoholic potassium hydroxide solution (25 g of KOH and 35 mL of sterile distilled water, brought to 100 mL with 100% ethanol). The mixture was vortexed for 2 min and then incubated for 3 h at 85 °C. Subsequently, 2 mL of sterile distilled water and 5 mL of n-heptane (Sigma-Aldrich, St. Louis, MO, USA) were added and vortexed for 5 min. The samples were kept at room temperature for 1 to 2 h to allow phase separation. The reading was performed using a spectrophotometer between 240 and 300 nm. 

The ergosterol content was calculated employing two equations:


(1)
value 1=[(A281.5 /290)×F]/wet cell weight



(2)
value 2=[(A230 /518)×F]/wet cell weight



(3)
ergosterol percentage=value 1−value 2


“F” represents the dilution factor in ethanol, while 290 and 518 are fixed values determined for the crystalline ergosterol and the dihydroergosterol, respectively. The analyses were performed in triplicate, and the statistical difference, evaluated by the Student’s *t*-test, was considered significant when *p* ≤ 0.05.

### 2.8. Quantification of Reactive Oxygen Species

The induction of oxidative stress caused in cells exposed to OPI was assessed by quantifying ROS using dichlorodihydrofluorescein 2,7-diacetate dye—DCFH-DA (Sigma-Aldrich, St. Louis, MO, USA). DCFH-DA is a non-fluorescent compound that easily permeates cell membranes. Once inside the cell, it is deacetylated by intracellular esterases, converting it into non-fluorescent dichlorodihydrofluorescein (DCFH). In the presence of ROS, such as hydrogen peroxide or hydroxyl radicals, DCFH undergoes oxidation, converting it into the highly fluorescent compound 2′,7′-dichlorofluorescein (DCF).

Cells, with or without OPI, were observed at time intervals of 9, 12, and 24 h. One mL of each sample was collected, washed twice with PBS 1×, and then resuspended in 500 μL of a solution containing 25 µM of dichlorodihydrofluorescein 2,7-diacetate dye (Sigma-Aldrich). After 30 min of incubation in the dark at room temperature, the samples were washed twice with PBS 1× and resuspended in 500 μL of PBS 1×. Fluorescence analysis was conducted in triplicate using a fluorescence microscope (Zeiss Axiocam MRc-Scope A1) with a wavelength of 490–516 nm. Pixel quantification using AxioVision 4.8 software (Carl Zeiss, Jena, Germany) analyzed all fluorescent and well-delimited cells. The statistics were evaluated by Student’s *t*-test and considered significant when *p* ≤ 0.05.

## 3. Results

### 3.1. Temporal Dynamics of the Viability of Paracoccidioides Exposed to OPI

We evaluated the antifungal efficacy of OPI by temporally monitoring the survival of *P. brasiliensis* after 72 h. The viability was significantly affected after 9 h. After 72 h, 97% cell death was observed (Figure 1). These results are important to determine the initial time of antifungal action and define the time for proteomic analysis to obtain precise information on the proteins differentially expressed in response to the treatment. Considering that for the time interval of 9 h we observed a cell viability of 94.7% in the treated condition (presence of OPI) and 97.6% in the control condition (absence of OPI), we chose this time to perform the proteomic analysis, since we observed a minimal decrease in cell viability in the treated condition in this period. This period represents a critical time when cells are responding to the stress caused by the treatment and may be activating adaptive metabolic pathways or remodeling their metabolism to adapt to new conditions. 

### 3.2. Comparative Proteomics Reveals a Profile of Proteins Differentially Modulated by P. brasiliensis

Comparative proteomics identified 362 proteins using nanoUPLC-MSE, and 178 proteins were differentially expressed. Among these, 52 were downregulated (Appendix A and Figure 2A), and 126 upregulated (Appendix A and Figure 2A). Additionally, 16 proteins were found exclusively in the treatment conditions, while 32 were found only in the control group. These proteins were classified based on their associated metabolic processes. Among the downregulated proteins (Figure 2C), the most representative classes were metabolism (19 proteins), energy (10 proteins), protein synthesis (7 proteins), and unclassified proteins (5 proteins). Among upregulated proteins (Figure 2D), the most representative functional categories were protein synthesis (46 proteins), metabolism (30 proteins) with a highlight on amino acid metabolism (20 proteins), unclassified proteins (19 proteins), and energy (18 proteins).

### 3.3. Antifungal OPI Induces Ethanol Production in P. brasiliensis

Once we observed the increase in the abundance of alcohol dehydrogenase 1, we performed the ethanol measurements in cells exposed to OPI and compared them to control cells. The results demonstrated an increase in ethanol production in cells treated with the compound (Figure 3). 

### 3.4. The Membrane Integrity and Composition Are Altered by OPI 

To verify possible alterations caused by OPI in the fungal cell membrane, an EPR assay and ergosterol quantification were performed. Parameter 2A_//_, measured directly from the EPR spectra, indicated an increase in the membrane rigidity after 9 h of treatment with this compound (Figure 4A), indicating that OPI may be affecting the integrity and structure of the fungal membrane. 

The ergosterol content, an essential component of the fungal membrane, showed a significant increase after 48 h (Figure 4B). These findings suggest that *P. brasiliensis* initially modifies the lipid composition of the membrane, leading to changes in membrane rigidity, even without an immediate increase in ergosterol production. However, after a prolonged period of exposure (48 h), *Paracoccidioides* increased ergosterol production.

### 3.5. OPI Increases Oxidative Stress in Cells

Superoxide dismutase, an important protein involved in defense against oxidative stress, showed increased abundance after exposure of *P. brasiliensis* to OPI. However, catalase was found to be downregulated. Thus, we investigated whether this compound could act as an inducer of ROS. After 24 h of incubation of the fungal cells with the compound, an increase in fluorescence was observed (Figure 5A,B), indicating an increase in ROS. Thus, the regulation of enzymes related to oxidative stress metabolism, as observed in the proteomic analysis, is consistent with the quantification of ROS.

## 4. Discussion

The data presented in the proteomic assays (Appendix A) indicate that cellular metabolic processes of the fungus *P. brasiliensis* underwent significant changes during exposure to OPI, suggesting a metabolic regulation in response to the antifungal. This regulation is strengthened by the upregulated proteins involved in pyruvate production, including hexokinase and pyruvate kinase. Furthermore, enzymes involved in the synthesis of threonine from aspartate (aspartate-semialdehyde dehydrogenase, threonine synthase, and L-threonine 3-dehydrogenase), degradation of glycine (glycine cleavage system T protein), degradation of methionine (S-adenosylmethionine synthase), and degradation of ornithine (ornithine aminotransferase and 1-pyrroline-5-carboxylate dehydrogenase) were also identified with an increased abundance. These enzymes can indirectly generate pyruvate as an end product of their metabolic pathways. 

Proteins from other pathways, such as the glyoxylate and methylcitrate cycles, were also induced by *P. brasiliensis* in response to OPI. The glyoxylate cycle serves as an alternative route to the TCA cycle, which shares the enzymatic activities of malate dehydrogenase, citrate synthase, and aconitase. However, replacing the two decarboxylation steps of the TCA cycle, the major enzymes of the glyoxylate cycle, malate synthase, and isocitrate lyase, convert isocitrate and acetyl-CoA into succinate and malate, which allows several microorganisms to grow using fatty acids and 2-carbon compounds as a carbon source [20]. During processes such as carbon privation [21] and the transition from mycelium to yeast [22], isocitrate lyase has been shown to be induced, highlighting its important role in the pathogenicity of the fungus. Our proteomic results revealed that in the presence of OPI, *P. brasiliensis* upregulated isocitrate lyase (Appendix A).

The methylcitrate cycle is another alternative metabolic pathway that plays an important role in the metabolism of propionyl-CoA, a byproduct of amino acids, odd-chain fatty acids, and certain intermediate metabolites. Accumulation of propionyl-CoA can be highly toxic to cells and needs to be metabolized [23]. Furthermore, through the methylcitrate cycle, propionyl-CoA can be converted to pyruvate. We observed an upregulation of 2-methylcitrate synthase in our proteomic analyses. These findings suggest that other pathways are also induced and lead to pyruvate production.

Pyruvate can be oxidized, resulting in the formation of acetyl-CoA or conversion to ethanol and CO_2_ via alcoholic fermentation. Since dihydrolipoyl dehydrogenase was induced, we believe that part of the pyruvate is being directed toward the production of acetyl-CoA. However, we observed that some proteins related to the TCA cycle, such as citrate synthase, the alpha subunit of succinyl-CoA ligase, and succinate dehydrogenase, were down-regulated. These proteins play a crucial role in the TCA cycle, which is responsible for energy production in the form of ATP. The negative regulation of these proteins suggests a reduction in TCA cycle activity. It is possible that acetyl-CoA is being channeled to other metabolic pathways as an adaptive response of the fungus to the compound. This redirection of metabolic resources could favor pathways that generate less energy but are more advantageous for the survival of the fungus under stress conditions.

The upregulation of the enzyme alcohol dehydrogenase suggests that alcoholic fermentation is taking place to produce ethanol, which is associated with virulence in fungi. During metabolic processes such as fermentation, the CO_2_ molecule is released. CO_2_ fixation occurs through the interconversion of CO_2_ to HCO_3_ catalyzed by carbonic anhydrases. This process not only regulates the adequate carbon supply for cellular metabolism but also contributes to various signaling processes in fungi, including morphology and communication [24,25]. Carbonic anhydrases have been linked to the development and virulence of *Candida albicans* and *Cryptococcus neoformans* [26]. In *P. brasiliensis*, carbonic anhydrases have been upregulated during the infectious process [22] and under iron starvation [27]. Thus, the upregulation of carbonic anhydrase in the presence of OPI (Appendix A) suggests that this enzyme may play a significant role in the adaptive response of *P. brasiliensis* to the antifungal, potentially contributing to its survival and virulence. It is known that alcoholic fermentation is an important process for fungal virulence and survival [28,29]. In *Paracoccidioides* spp., ethanol production is regulated under different conditions, such as carbon deprivation [21] and in the presence of argentilactone, a natural antifungal that promotes oxidative stress [30]. Therefore, the ethanol production by *Paracoccidioides* in response to antifungal OPI may occur as an adaptative and survival strategy against the stress caused by the treatment.

The proteomic analyses also revealed the upregulation of acetyl-CoA acetyltransferase. This enzyme catalyzes the conversion of acetyl CoA to acetoacetyl-CoA, which is an important metabolite in ergosterol biosynthesis. These findings suggest that in the presence of OPI, *P. brasiliensis* induces metabolic pathways for the production of acetyl-CoA, directing this molecule toward the synthesis of ergosterol. A similar mechanism was proposed in *P. lutzzii* in the presence of itraconazole [31], which showed that *P. lutzii* upregulated the expression of genes involved in various metabolic pathways that converge on the production of acetyl-CoA, also utilizing this molecule in the synthesis of ergosterol. 

The sterol content is crucial for the fluidity and permeability of biological membranes, osmotic stress tolerance, and the development and survival of fungi [32]. Erg proteins are a large family of proteins involved in the biosynthesis pathway of ergosterol, which is necessary for maintaining cell membrane integrity. Consequently, erg proteins have been targeted by antifungal agents such as azoles [33]. Proteins associated with sterols biosynthesis, including protein Erg 28, farnesyl pyrophosphate synthase, and fatty acid synthase subunits beta and alpha, were downregulated (Appendix A). The ERG gene expression was also downregulated in *C. albicans* in response to farnesol, a compound that influences the integrity of the plasmatic membrane [34]. Thus, the upregulation of these proteins can indicate a potential disruption in the biosynthetic pathway of sterols and, consequently, cell membrane integrity alteration. This hypothesis can be supported by the results of electron paramagnetic resonance, which demonstrated an increase in membrane rigidity in the membrane of *P. brasiliensis*. We observed an increase in the ergosterol content in cells exposed to the antifungal after 48 h, which can be explained by compensatory mechanisms or alternative regulatory pathways that occur in response to the interruption of the sterol synthesis pathway. However, we believe that this change was not sufficient for membrane restoration, as the viability of *P. brasiliensis* continued to decrease over time.

The increase in ROS formation after exposure to OPI suggests that the compound may have direct or indirect effects on cellular redox balance. The compound itself may possess chemical properties that promote ROS generation, or alternatively, the increased ROS may be due to the activation of specific metabolic pathways leading to ROS production. The downregulation of catalase, an enzyme involved in hydrogen peroxide detoxification, and the upregulation of superoxide dismutase, responsible for converting superoxide to hydrogen peroxide and oxygen, may indicate the existence of regulatory mechanisms aimed at maintaining redox balance within the fungal cells. Under conditions of oxidative stress, the regulation of these enzymes may be coordinated to activate antioxidant defense pathways and limit oxidative damage, as shown by Araújo et al. [30]. That study also found that *P. brasiliensis* upregulates superoxide dismutase in the presence of argentilactone, an antifungal inducer of ROS. 

Based on the observed results, we outline an overview of the metabolic alterations of *P. brasiliensis* in the presence of OPI (Figure 6).

In the context of drug discovery and the proposition of molecular targets for compounds, proteomics has been recognized as a valuable tool. However, it is important to acknowledge certain limitations associated with its application in this specific context. Many proteins remain unidentified or have undefined functions in proteomic databases, thereby limiting our ability to identify relevant molecular targets for specific compounds. For instance, in our proteome, several proteins lack defined functions in the databases, leading to their exclusion from the construction of the metabolic changes model. Additionally, the variation in protein expression across different biological contexts requires analysis under diverse conditions to identify more robust targets. Moreover, the interaction of proteins with cofactors and protein complexes may be overlooked in proteomic analysis, affecting the accuracy of target identification. One strategy to overcome these limitations is the integration of omics technologies, such as genomics, transcriptomics, interactomics, and metabolomics, which can provide a more comprehensive view of biological mechanisms. This integration has already been employed, as demonstrated in the case of the argentilactone compound [30,35,36].

## 5. Conclusions

Through proteomic analyses, we identified several proteins that were differentially regulated, indicating a metabolic reorganization of the fungus in the presence of OPI. We observed a significant increase in ethanol production, suggesting that the activation of alcoholic fermentation serves as a survival strategy for the fungus in response to the stress induced by the treatment. Additionally, we observed an increase in membrane rigidity, suggesting a potential impact of the compound on the structural integrity of the fungal membrane. Moreover, prolonged exposure to the compound led to an elevation in the ergosterol content, which is a vital component of the membrane. This finding suggests that the fungus may undergo adaptations aimed at restoring the normal lipid composition of the membrane. Furthermore, our quantification of ROS demonstrated an augmentation in oxidative stress within cells exposed to OPI, indicating an adaptive response of the fungus to the treatment.

## Figures and Tables

**Figure 1 jof-09-00854-f001:**
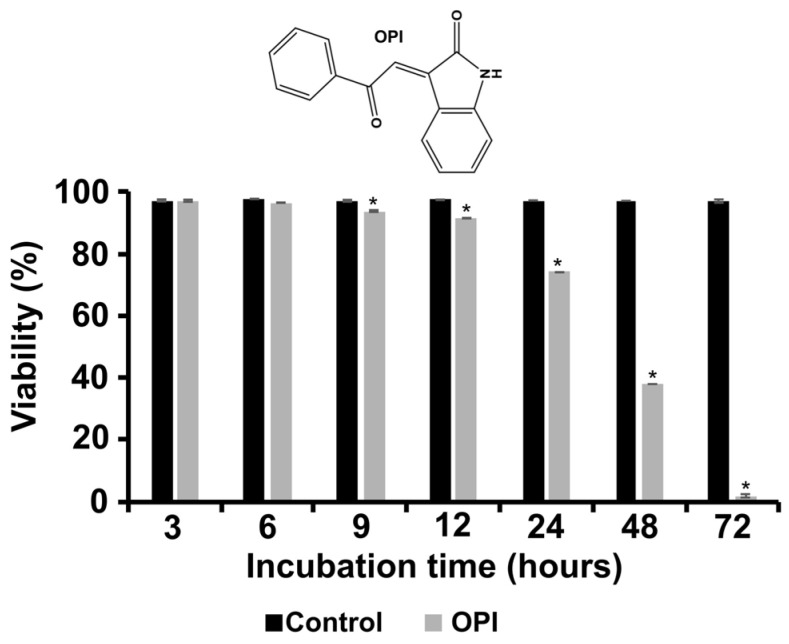
Monitoring the temporal viability of *P. brasiliensis* in the presence of the compound OPI. Fungal cells were exposed to the OPI compound for a specific period of time, followed by the evaluation of cellular viability over time. A gradual reduction in fungal cell viability is observed as the duration of exposure to the OPI compound increases, indicating its potential inhibitory effect on the growth of *P. brasiliensis*. * *p* < 0.05.

**Figure 2 jof-09-00854-f002:**
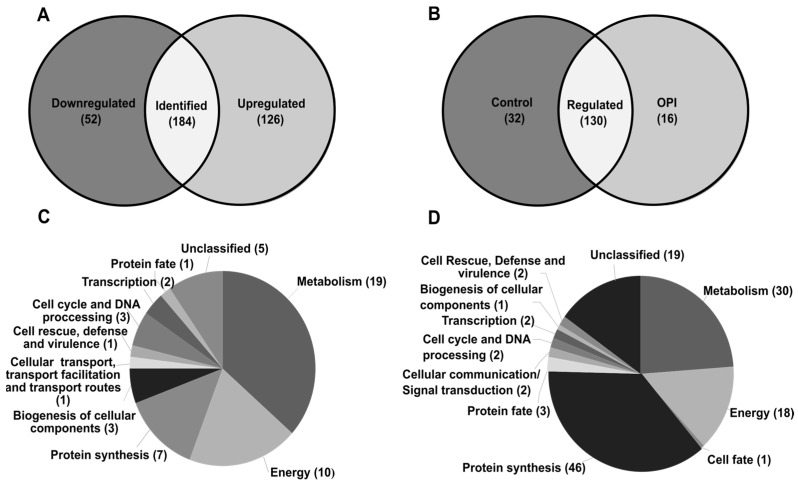
Distribution of identified and differentially expressed proteins in *P. brasiliensis* by metabolic processes after exposure to OPI. The Venn diagram shows the number of up and downregulated proteins in relation to the proteins identified in the proteome (**A**) and the number of proteins exclusively identified in the control condition or in the presence of OPI (**B**). The pie charts provide an overview of the distribution of proteins based on their metabolic processes. Downregulated proteins categorized by metabolic processes (**C**), and upregulated proteins categorized by metabolic processes (**D**).

**Figure 3 jof-09-00854-f003:**
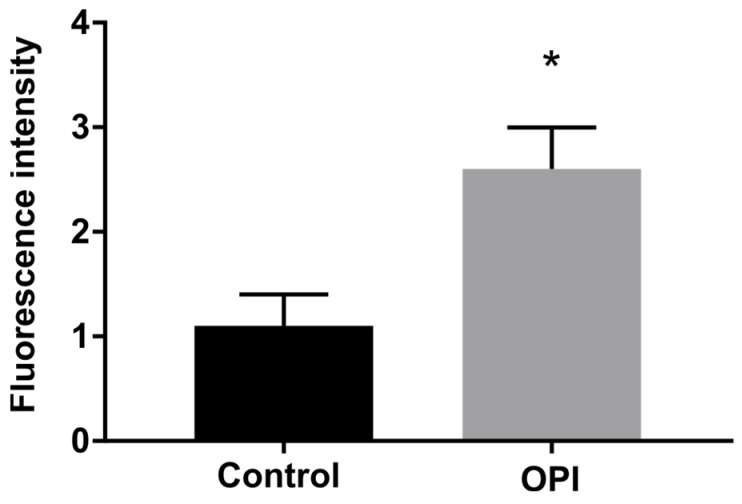
Ethanol concentration produced by cells exposed to OPI. The graph displays the ethanol content measured in cells after exposure to OPI. * indicates a significant difference between the control and treated samples, with a *p*-value of ≤ 0.05. The error bars represent the standard deviation between biological triplicates. Notably, a pronounced increase in ethanol production was observed in cells treated with the OPI compound, indicating its potential impact on the cellular metabolic pathways involved in ethanol synthesis. This finding suggests that OPI treatment may influence the metabolic processes leading to increased ethanol production in the cells under investigation.

**Figure 4 jof-09-00854-f004:**
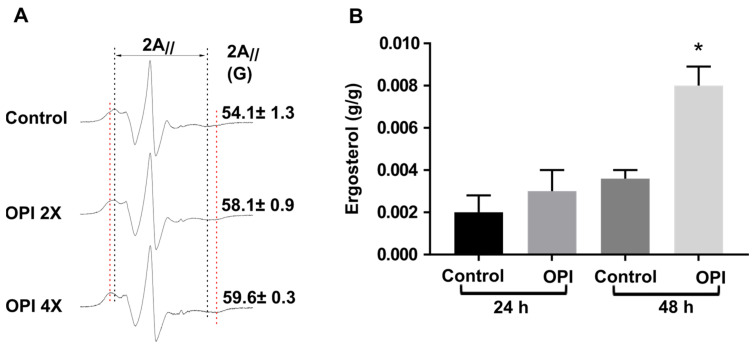
Changes in membrane and ergosterol production caused by exposure to OPI. (**A**) Representative EPR spectra of spin probe 5-DSA in fungal membranes for the untreated samples (control) and samples treated with OPI at concentrations of 2× and 4× in culture medium. The mean ± S.D. values of the EPR parameter 2A_//_ (outer hyperfine splitting), which is given by the separation in magnetic field units between the first peak and the last inverted peak of the spectrum, are indicated. The increase in 2A_//_ indicates membrane rigidity. The EPR spectra were recorded with a scan range of the total magnetic field of 100 G (X-axis) and intensity represented in arbitrary units (Y-axis). (**B**) Concentration of intracellular ergosterol in samples exposed or not to OPI for 24 and 48 h. The * indicates a significant difference between the control and treated samples, with a *p*-value ≤ 0.05. The error bar represents the deviation between the biological triplicate.

**Figure 5 jof-09-00854-f005:**
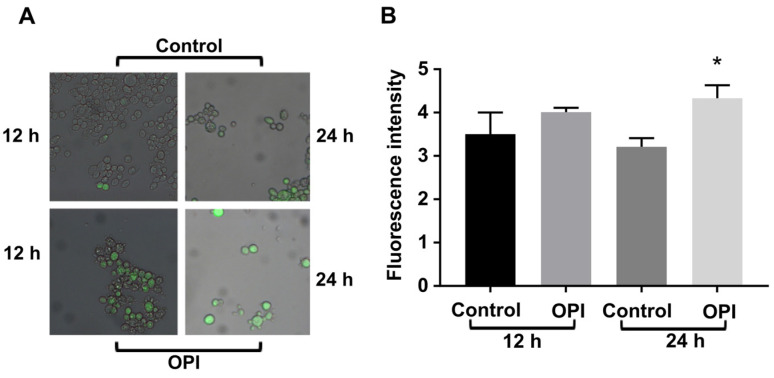
Induction of oxidative stress in cells exposed to OPI. (**A**) Fluorescence microscopy using DCFH-DC to quantify reactive oxygen species (ROS) in cells exposed or not to the compound for 12 and 24 h. Image obtained at a magnification of 40× (**B**) Quantification of ROS. The fluorescence intensity (in pixels) of the stained cells was measured. Error bars represent the standard deviation of three biological replicates. The statistical significance between the control and treated samples was determined using Student’s *t*-test. An increase in ROS levels was observed when the fungal cells were exposed to the OPI compound. The fluorescence microscopy images in panel (**A**) demonstrate a higher intensity of fluorescence, indicating elevated ROS production in cells treated with OPI compared to the control. These findings, supported by the quantification results in panel (**B**), underscore the ability of the OPI compound to induce oxidative stress in the fungal cells. The * indicates a significant difference between the control and treated samples, with a *p*-value ≤ 0.05.

**Figure 6 jof-09-00854-f006:**
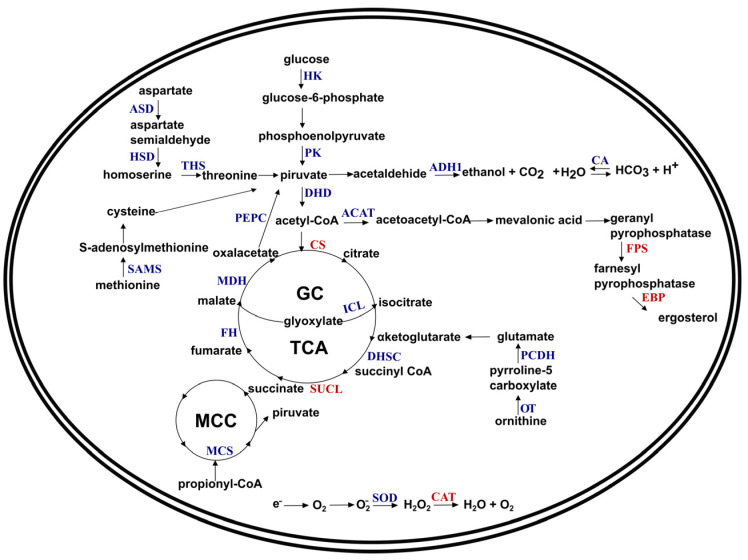
Overview of the metabolic changes of *P. brasiliensis* in the presence of OPI. The upregulated proteins are highlighted in blue, and the downregulated are highlighted in red. ASD—aspartate-semialdehyde dehydrogenase, HSD—homoserine dehydrogenase, THS—threonine synthase, HK—hexokinase, PGK—phosphoglycerate kinase, PGM—phosphoglycerate mutase, PK—pyruvate kinase, DHD—dihydrolipoyl dehydrogenase, PEPC—phosphoenolpyruvate carboxykinase, SAMS—S-adenosylmethionine synthase, ADH1—alcohol dehydrogenase 1, CA—carbonic anhydrase, ACAT—acetyl-CoA acetyltransferase, FAS—fatty acid synthase, FPS—farnesyl pyrophosphate synthetase, EBP—ergosterol biosynthesis protein Erg 28, CS—citrate synthase, MDH—malate dehydrogenase, FH—fumarate hydratase, SDH—succinate dehydrogenase, SUCL—succinyl-CoA ligase, DHSC—dihydrolipoamide succinyltransferase, ICL—isocitrate lyase, MCS—2-methylcitrate synthase, PDCDH—1-pyrroline-5-carboxylate dehydrogenase, OT—ornithine aminotransferase, SOD-superoxide dismutase, CAT—Catalase, MCC—methyl citrate cycle, GC—glyoxylate cycle, TCA—tricarboxylic acid cycle.

## Data Availability

The submission of the proteomics results was assigned the identifier PASS05839 in the PeptideAtlas repository.

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
