# Peer review of "Proteomic Profiling of Paracoccidioides brasiliensis in Response to Phenacylideneoxindol Derivative: Unveiling Molecular Targets and Pathways"

_jof, 2023, doi:10.3390/jof9080854_

Round 1
Reviewer 1 Report
The paper is easy to follow, although there are 3 major issues:
1. Fold changes below and above 2 seem insignificant in proteomic studies and the cut-off point 1.3 speaks of very small changes. More importantly, the lack of statistics with such small differences disqualifies these data as valid. It cannot be ruled out that if a statistical analysis were carried out, it would turn out that most of the differences are not significant. Authors should necessarily provide statistical analysis. In addition, differences of the order of max. 2.3- fold raise the question of whether the Authors performed MS analyzes at the right time after OPI induction. OPI definitely affects the functioning of cells, hence the observed changes seem to be inadequately small. Have the Authors deposited the original MS data? It would be worth presenting a list of proteins in the supplement, the level of which did not change.
2. One of the conclusions of this work is that OPI induces changes in membrane structure. To make the work more complete, a proteomic analysis of membrane proteins would be useful to check whether their amount changes under the influence of OPI.
3. Ergosterol production- Figure 4b shows a greater than 50% increase in membrane ergosterol in cells treated with OPI for 48 hours. Only figure 1 shows that at 48 hours more than half of the cells treated with OPI are dead. Were the dead cells separated from the dead ones? If not, then the conclusion “However, after a prolonged period of exposure, Paracoccidioides may develop adaptations to counteract the effects of OPI on the membrane, such as increased ergosterol production which may contribute to the restoration of the membrane’s normal lipid composition” does not necessarily have to be accurate because it is not known whether obtained data refer to living or dead cells.
Minor issues:
4. Line 150: these 3 replicates are technical replicates after digestion or biological replicates? If after digestion, how many biological replicates were there or from how many mixed biological replicates were the proteins for MS isolated?
5. Line 159: the Authors exposed cells to OPI, not curcumin.
6. Lines 166-167: Was it biological or technical replicate?
7. Line 211: what is the working principle of this reagent?
8. Line 222: Is the viability changed significantly from 12 or 9 hours? Because the statistically significant result is marked on the graph from the 9th hour of incubation with OPI.
9. Line 270: what are the proteins? Because in the table, only one protein alcohol dehydrogenase 1 catches the eye.
Author Response
The paper is easy to follow, although there are 3 major issues:
- Fold changes below and above 2 seem insignificant in proteomic studies and the cut-off point 1.3 speaks of very small changes. More importantly, the lack of statistics with such small differences disqualifies these data as valid. It cannot be ruled out that if a statistical analysis were carried out, it would turn out that most of the differences are not significant.
Authors should necessarily provide statistical analysis. In addition, differences of the order of max. 2.3- fold raise the question of whether the Authors performed MS analyzes at the right time after OPI induction. OPI definitely affects the functioning of cells, hence the observed changes seem to be inadequately small.
Have the Authors deposited the original MS data?
It would be worth presenting a list of proteins in the supplement, the level of which did not change.
Response:
Thank you for your feedback on the fold changes observed in our proteomic study.
We acknowledge that in proteomic studies, fold changes below 2 may generally be considered less significant. However, it is worth noting that a cut-off point of 1.3 has been used in other research articles and can hold relevance in a broader context.
Here are some examples of recent studies that have used a fold change of 1.3 as a cut-off point in proteomic analyses.
- Silva L.C. et al. Proteomic Response of Paracoccidioides brasiliensisExposed to the Antifungal 4-Methoxynaphthalene-N-acylhydrazone Reveals Alteration in Metabolism. J. fungi. doi: 10.3390/jof9010066.
- Marcos Abc Júnior M.A et al. Proteomic identification of metabolic changes in Paracoccidioides brasiliensisinduced by a nitroheteroarylchalcone. Future Microbiol. doi: 10.2217/fmb-2022-0150.
- Lan J. et al. Quantitative Proteomic Analysis Uncovers the Mediation of Endoplasmic Reticulum Stress-Induced Autophagy in DHAV-1-Infected DEF Cells. 2019. Int J Mol Sci. doi: 3390/ijms20246160.
Taking into consideration your remarks about the need to provide a statistical analysis, we would like to clarify that during the data processing of our mass spectrometry analyses, we used ProteinLynx Global Server version 2 software. This software generates a statistical analysis, which we employed as a protocol for the initial classification of induced or repressed proteins. In this protocol, we consider proteins as repressed when their values fall within the range of 0 to 0.5 and as induced when they are within the range of 0.95 to 1. Additionally, we also applied the classification by fold change to reinforce the analysis. We have added the statistical information in Supplementary Table 1 and 2.
Regarding the time of 9 hours chosen for the MS analyses, we believe it is appropriate because we observed only a slight impact of the OPI compound on cell viability at this time point. Longer incubation time intervals, which could result in lower cell viability, might hinder our ability to understand the initial response of the fungus to the compound. Additionally, longer incubation times could lead to the analysis of cellular death responses, which we aim to avoid in this particular study.
We deposited the original MS data. The submission of the proteomics’ results was assigned the identifier PASS05839 in the PeptideAtlas repository, and this information has been added to the methodology section.
The list of proteins which did not change was included as a supplementary table S3.
- One of the conclusions of this work is that OPI induces changes in membrane structure. To make the work more complete, a proteomic analysis of membrane proteins would be useful to check whether their amount changes under the influence of OPI.
Response: We sincerely appreciate your valuable comment and suggestion regarding the inclusion of a proteomic analysis to investigate changes in membrane protein quantities under the influence of OPI in our work. However, a proteomic analysis is a complex task that requires proper planning, resources, and substantial time to be executed. Due to the scope of this article and other limitations, it is not feasible to conduct this specific analysis at the moment. Nonetheless, we will take this recommendation into consideration for future studies.
- Ergosterol production- Figure 4b shows a greater than 50% increase in membrane ergosterol in cells treated with OPI for 48 hours. Only figure 1 shows that at 48 hours more than half of the cells treated with OPI are dead. Were the dead cells separated from the dead ones? If not, then the conclusion “However, after a prolonged period of exposure, Paracoccidioides may develop adaptations to counteract the effects of OPI on the membrane, such as increased ergosterol production which may contribute to the restoration of the membrane’s normal lipid composition” does not necessarily have to be accurate because it is not known whether obtained data refer to living or dead cells.
Response: Considering the limitation of the available data and the lack of distinction between living and dead cells after 48 hours of OPI treatment, we acknowledge that the original conclusion might be premature in stating that Paracoccidioides may develop adaptations to counteract the effects of OPI on the membrane, such as increased ergosterol production. Therefore, we have made the suggested modification to the sentence in line 304: "However, after a prolonged period of exposure, Paracoccidioides increased ergosterol production which may contribute to the restoration of the membrane's normal lipid composition."
Minor issues:
- Line 150: these 3 replicates are technical replicates after digestion or biological replicates? If after digestion, how many biological replicates were there or from how many mixed biological replicates were the proteins for MS isolated?
Response: We performed biological triplicates in our study. The fungus Paracoccidioides brasiliensis was incubated in the presence and absence of the compound on different days. After obtaining the triplicates, the samples were pooled and subsequently digested. The resulting pool was analyzed in triplicate in the mass spectrometry analyses.
- Line 159: the Authors exposed cells to OPI, not curcumin.
Response: Thank you for bringing this to our attention. We have made the necessary correction in the article.
- Lines 166-167: Was it biological or technical replicate?
Response: The experiments were performed in triplicate, and the graph represents the average of these triplicates.
- Line 211: what is the working principle of this reagent?
Response: The working principle of dichlorodihydrofluorescein 2,7-diacetate (DCFH-DA) dye is based on its ability to detect reactive oxygen species (ROS) within cells. DCFH-DA is a non-fluorescent compound that easily permeates cell membranes. Once inside the cell, it is deacetylated by intracellular esterases, converting it into the non-fluorescent dichlorodihydrofluorescein (DCFH). In the presence of ROS, such as hydrogen peroxide or hydroxyl radicals, DCFH undergoes oxidation, converting it into the highly fluorescent compound 2',7'-dichlorofluorescein (DCF). The intensity of the fluorescence is directly proportional to the amount of ROS present in the cell, allowing researchers to quantify the level of oxidative stress or assess ROS production in various experimental conditions.
- Line 222: Is the viability changed significantly from 12 or 9 hours? Because the statistically significant result is marked on the graph from the 9th hour of incubation with OPI.
Response: Thank you for your question. Indeed, we observed a statistically significant change in viability starting from the 9th hour of incubation with OPI. Therefore, we chose this time point for the proteomic assays as it marked the initial detection of significant antifungal activity. By selecting this specific time point, we aimed to detect the early molecular events associated with the observed antifungal effects of OPI.
- Line 270: what are the proteins? Because in the table, only one protein alcohol dehydrogenase 1 catches the eye.
Response: Thank you for the correction. In line 270 the phrase “Once we observed the increase in the abundance of proteins involved in ethanol production, we performed the ethanol measurements in cells exposed to OPI and compared them to control cells” was changed to “Once we observed the increase in the abundance of alcohol dehydrogenase 1, we performed the ethanol measurements in cells exposed to OPI and compared them to control cells.
Reviewer 2 Report
This study investigates the proteomic and metabolic response of P. paracoccidiodes to a Phenacylideneoxindol derivative. This study is helpful in providing foundational knowledge on this organism and potential value of this antimicrobial agent. It was well written and easy to interpret. No major concerns.
Author Response
Thank you for your positive feedback on our study investigating the proteomic and metabolic response of P. paracoccidiodes to a Phenacylideneoxindol derivative.
Reviewer 3 Report
In this manuscript, number jof-2498666, entitled: “Proteomic profiling of Paracoccidioides brasiliensis in response to phenacylideneoxindol derivative: unveiling molecular targets and pathways” by do Carmo Silva et al., they performed a proteomic analysis on P. brasiliensis in response to (E)-3-(2-oxo-2-phenylethylidene)indolin-2-one (OPI) (which exhibits antifungal activity) and identified several proteins that were differentially regulated. These results indicate a metabolic reorganization of the fungus in the presence of OPI. Also, they observed an increase in etanol production, alterations in cell membrane, and ROS production in P. brasiliensis in response to OPI. These onservations help to understand the action mechanism of this compound which is considered a potentional candidate for the treatment of paracoccidioidomycosis.
Here are the review points:
1) The total number of identified proteins was low. You must considered to break fungal cell with liquid nitrogen. Based on various reports, cell disruption with liquid nitrogen is better for proteomic analysis. Please, include the reference of the method used for cell disruption. Please, include the protease inhibitor used.
2) Please, include the figure of the SDS-PAGE gel to assess the integrity of the proteins (supplementary figure).
3) In line 159 of Material and Methods section, “The proteins were later classified as induced or repressed after exposure to curcumin”, is curcumin correct? Please clarify.
4) In the Results section, include a figure that shows the total number of proteins identified and another figure with a Venn diagram that indicate those that were expressed, over-expressed, and exclusive in both conditions. In addition, include a table that shows the differentially expressed proteins associated with the integrity of the ethanol production, cell membrane, and ROS production.
5) Why were the ROS production assays done at 12 h instead of 9 h, considering that the proteomic analysis was carried out at 9 h? or, Why was it not considered to do the proteomic analysis at 12 h instead of 9 h?
6) In the Discussion section, please include the limitations of used methods and how to improve the proteomic analysis.
7) In all writing, please abbreviate hours, minutes, and seconds.
Minor editing of English language required.
Author Response
In this manuscript, number jof-2498666, entitled: “Proteomic profiling of Paracoccidioides brasiliensis in response to phenacylideneoxindol derivative: unveiling molecular targets and pathways” by do Carmo Silva et al., they performed a proteomic analysis on P. brasiliensis in response to (E)-3-(2-oxo-2-phenylethylidene)indolin-2-one (OPI) (which exhibits antifungal activity) and identified several proteins that were differentially regulated. These results indicate a metabolic reorganization of the fungus in the presence of OPI. Also, they observed an increase in etanol production, alterations in cell membrane, and ROS production in P. brasiliensis in response to OPI. These onservations help to understand the action mechanism of this compound which is considered a potentional candidate for the treatment of paracoccidioidomycosis.
Here are the review points:
1. The total number of identified proteins was low. You must considered to break fungal cell with liquid nitrogen. Based on various reports, cell disruption with liquid nitrogen is better for proteomic analysis. Please, include the reference of the method used for cell disruption. Please, include the protease inhibitor used.
Response: We appreciate your comments and suggestions regarding the total number of proteins identified in our study. We believe that the quantity of identified proteins is not directly associated with the lysis method. The lysis approach used in this article was chosen considering the standardized methodology in our laboratory and is also described in other studies that have shown good performance.
The reference to the extraction method was included in the manuscript at line 135.
The protease inhibitor used was included in Material and Methods.
3. Please, include the figure of the SDS-PAGE gel to assess the integrity of the proteins (supplementary figure).
Response: The SDS-PAGE gel was included as supplementary figure S1.
4. In line 159 of Material and Methods section, “The proteins were later classified as induced or repressed after exposure to curcumin”, is curcumin correct? Please clarify.
Response: The information was mistakenly included. We changed it to “The proteins were later classified as induced or repressed after exposure to OPI.”
5. In the Results section, include a figure that shows the total number of proteins identified and another figure with a Venn diagram that indicate those that were expressed, over-expressed, and exclusive in both conditions. In addition, include a table that shows the differentially expressed proteins associated with the integrity of the ethanol production, cell membrane, and ROS production.
Response: Thank you for your valuable feedback. We have addressed your suggestions accordingly. In the Results section, we have included the figures as requested. The table that shows the differentially expressed proteins associated with the integrity of the ethanol production, cell membrane, and ROS production was included as supplementary table S4.
6. Why were the ROS production assays done at 12 h instead of 9 h, considering that the proteomic analysis was carried out at 9 h? or, Why was it not considered to do the proteomic analysis at 12 h instead of 9 h?
Response: We conducted ROS production assay at 9, 12 and 24 hours.After analysis, we observed that the results at 9 hours were similar to those at 12 hours. However, when building the figure, we chose to add only 12 and 24 hours.
7. In the Discussion section, please include the limitations of used methods and how to improve the proteomic analysis.
Response: As requested, in the Discussion section of the article, the limitations of the methods used and strategies to improve proteomic analysis have been included. This information has been added to the text at Line 493.
8. In all writing, please abbreviate hours, minutes, and seconds.
Response: The changes have been made as requested.
Reviewer 4 Report
The authors previously identified antifungal activity of phenacylideneoxindoles against the important, understudied pathogen Paracoccidioides brasiliensis. In this manuscript, the authors report experiments aimed at determining mechanism or target for this effect. This is a clinically relevant topic, and identification of new antifungals, especially those with new targets, would be highly significant. They show pleomorphic relevant changes associated with phenacylideneoxindole treatment, including extensive proteomic changes somewhat clustered in particular metabolic pathways, increased ethanol production, changes in membrane fluidity and ergosterol content, and altered production of reactive oxidative species. The goals of the study and the experimental approach are reasonable. The phenotypic assays performed were generally driven by the proteomic findings. The presented results support the potential for phenacylideneoxindoles as antifungal agents as well as demonstrating a variety of important biological effects consistent with the antifungal activity. As such, the authors’ interpretations and conclusions are generally supported and valid. The study remains somewhat on a descriptive level, and the authors have not identified “THE mechanism” or “THE target” for phenacylideneoxindoles, but work significantly contributes to mechanistic understanding for the antifungal activity.
Specific comment: In lines 61-62, change “which that inhibit” to “which inhibits”
Author Response
The authors previously identified antifungal activity of phenacylideneoxindoles against the important, understudied pathogen Paracoccidioides brasiliensis. In this manuscript, the authors report experiments aimed at determining mechanism or target for this effect. This is a clinically relevant topic, and identification of new antifungals, especially those with new targets, would be highly significant. They show pleomorphic relevant changes associated with phenacylideneoxindole treatment, including extensive proteomic changes somewhat clustered in particular metabolic pathways, increased ethanol production, changes in membrane fluidity and ergosterol content, and altered production of reactive oxidative species. The goals of the study and the experimental approach are reasonable. The phenotypic assays performed were generally driven by the proteomic findings. The presented results support the potential for phenacylideneoxindoles as antifungal agents as well as demonstrating a variety of important biological effects consistent with the antifungal activity. As such, the authors’ interpretations and conclusions are generally supported and valid. The study remains somewhat on a descriptive level, and the authors have not identified “THE mechanism” or “THE target” for phenacylideneoxindoles, but work significantly contributes to mechanistic understanding for the antifungal activity.
Specific comment: In lines 61-62, change “which that inhibit” to “which inhibits”
Response: Thank you for the correction. We have made the necessary correction as per your recommendation. The phrase "which that inhibit" has been revised to "which inhibits" in lines 61-62 of the article.
Round 2
Reviewer 1 Report
The paper is easy to follow, although there are 3 major issues:
-
Fold changes below and above 2 seem insignificant in proteomic studies and the cut-off point 1.3 speaks of very small changes. More importantly, the lack of statistics with such small differences disqualifies these data as valid. It cannot be ruled out that if a statistical analysis were carried out, it would turn out that most of the differences are not significant.
Authors should necessarily provide statistical analysis. In addition, differences of the order of max. 2.3- fold raise the question of whether the Authors performed MS analyzes at the right time after OPI induction. OPI definitely affects the functioning of cells, hence the observed changes seem to be inadequately small.
Have the Authors deposited the original MS data?
It would be worth presenting a list of proteins in the supplement, the level of which did not change.
Response:
Thank you for your feedback on the fold changes observed in our proteomic study.
We acknowledge that in proteomic studies, fold changes below 2 may generally be considered less significant. However, it is worth noting that a cut-off point of 1.3 has been used in other research articles and can hold relevance in a broader context.
Here are some examples of recent studies that have used a fold change of 1.3 as a cut-off point in proteomic analyses.
-
Silva L.C. et al. Proteomic Response of Paracoccidioides brasiliensisExposed to the Antifungal 4-Methoxynaphthalene-N-acylhydrazone Reveals Alteration in Metabolism. J. fungi. doi: 10.3390/jof9010066.
-
Marcos Abc Júnior M.A et al. Proteomic identification of metabolic changes in Paracoccidioides brasiliensisinduced by a nitroheteroarylchalcone. Future Microbiol. doi: 10.2217/fmb-2022-0150.
-
Lan J. et al. Quantitative Proteomic Analysis Uncovers the Mediation of Endoplasmic Reticulum Stress-Induced Autophagy in DHAV-1-Infected DEF Cells. 2019. Int J Mol Sci. doi: 3390/ijms20246160.
Of course, there are publications showing such small differences, but it still rather means that the experiment was poorly designed. In addition, 2 of the 3 papers listed are from the same team as the Authors. Such small changes indicate a small effect of the substance on the proteome, which is not in line with the other results. Also, the third paper cited by Lan shows a level of statistical significance - your paper does not. For me, these results are still not credible.
Taking into consideration your remarks about the need to provide a statistical analysis, we would like to clarify that during the data processing of our mass spectrometry analyses, we used ProteinLynx Global Server version 2 software. This software generates a statistical analysis, which we employed as a protocol for the initial classification of induced or repressed proteins. In this protocol, we consider proteins as repressed when their values fall within the range of 0 to 0.5 and as induced when they are within the range of 0.95 to 1. Additionally, we also applied the classification by fold change to reinforce the analysis. We have added the statistical information in Supplementary Table 1 and 2.
A protocol saying that, for example, repressed proteins have fold values in the range of 0 to 0.5 is not a statistic! It’s just an estimation of fold change. In tables 1 and 2 from the supplement there is not a word about statistics. What was the p value? If the software perfomed statistical analysis, what was the number of proteins for which the fold change was not statistically significant? Normally there is a plenty of identified proteins in such analysis, but the significant change is only for some of them. The presented data is still unreliable without presented statistical analysis. If the Authors cannot say how was statistical analysis performed then they should use different tool to perform statistical analysis.
Regarding the time of 9 hours chosen for the MS analyses, we believe it is appropriate because we observed only a slight impact of the OPI compound on cell viability at this time point. Longer incubation time intervals, which could result in lower cell viability, might hinder our ability to understand the initial response of the fungus to the compound. Additionally, longer incubation times could lead to the analysis of cellular death responses, which we aim to avoid in this particular study.
We deposited the original MS data. The submission of the proteomics’ results was assigned the identifier PASS05839 in the PeptideAtlas repository, and this information has been added to the methodology section.
OK
The list of proteins which did not change was included as a supplementary table S3.
OK
-
One of the conclusions of this work is that OPI induces changes in membrane structure. To make the work more complete, a proteomic analysis of membrane proteins would be useful to check whether their amount changes under the influence of OPI.
Response: We sincerely appreciate your valuable comment and suggestion regarding the inclusion of a proteomic analysis to investigate changes in membrane protein quantities under the influence of OPI in our work. However, a proteomic analysis is a complex task that requires proper planning, resources, and substantial time to be executed. Due to the scope of this article and other limitations, it is not feasible to conduct this specific analysis at the moment. Nonetheless, we will take this recommendation into consideration for future studies.
I still think that this experiment should be included if the Authors want to “sell” their work in good journal.
-
Ergosterol production- Figure 4b shows a greater than 50% increase in membrane ergosterol in cells treated with OPI for 48 hours. Only figure 1 shows that at 48 hours more than half of the cells treated with OPI are dead. Were the dead cells separated from the dead ones? If not, then the conclusion “However, after a prolonged period of exposure, Paracoccidioides may develop adaptations to counteract the effects of OPI on the membrane, such as increased ergosterol production which may contribute to the restoration of the membrane’s normal lipid composition” does not necessarily have to be accurate because it is not known whether obtained data refer to living or dead cells.
Response: Considering the limitation of the available data and the lack of distinction between living and dead cells after 48 hours of OPI treatment, we acknowledge that the original conclusion might be premature in stating that Paracoccidioides may develop adaptations to counteract the effects of OPI on the membrane, such as increased ergosterol production. Therefore, we have made the suggested modification to the sentence in line 304: "However, after a prolonged period of exposure, Paracoccidioides increased ergosterol production which may contribute to the restoration of the membrane's normal lipid composition."
It sounds a bit better, but still we don’t know if the higher ergosterol production may contribute to the restoration of the mebrane normal lipid composition, because it still seems that ergosterol level is corelated with the percentage of dead cells- so, higher ergosterol didn’t rescue the cells form death, hence probably yhe restoration was a failure.
Minor issues:
-
Line 150: these 3 replicates are technical replicates after digestion or biological replicates? If after digestion, how many biological replicates were there or from how many mixed biological replicates were the proteins for MS isolated?
Response: We performed biological triplicates in our study. The fungus Paracoccidioides brasiliensis was incubated in the presence and absence of the compound on different days. After obtaining the triplicates, the samples were pooled and subsequently digested. The resulting pool was analyzed in triplicate in the mass spectrometry analyses.
This information should be incorporated into the text
-
Line 159: the Authors exposed cells to OPI, not curcumin.
Response: Thank you for bringing this to our attention. We have made the necessary correction in the article.
OK
-
Lines 166-167: Was it biological or technical replicate?
Response: The experiments were performed in triplicate, and the graph represents the average of these triplicates.
OK
-
Line 211: what is the working principle of this reagent?
Response: The working principle of dichlorodihydrofluorescein 2,7-diacetate (DCFH-DA) dye is based on its ability to detect reactive oxygen species (ROS) within cells. DCFH-DA is a non-fluorescent compound that easily permeates cell membranes. Once inside the cell, it is deacetylated by intracellular esterases, converting it into the non-fluorescent dichlorodihydrofluorescein (DCFH). In the presence of ROS, such as hydrogen peroxide or hydroxyl radicals, DCFH undergoes oxidation, converting it into the highly fluorescent compound 2',7'-dichlorofluorescein (DCF). The intensity of the fluorescence is directly proportional to the amount of ROS present in the cell, allowing researchers to quantify the level of oxidative stress or assess ROS production in various experimental conditions.
I know that- the readers may not, so the summary of it should be pointed in the text
-
Line 222: Is the viability changed significantly from 12 or 9 hours? Because the statistically significant result is marked on the graph from the 9th hour of incubation with OPI.
Response: Thank you for your question. Indeed, we observed a statistically significant change in viability starting from the 9th hour of incubation with OPI. Therefore, we chose this time point for the proteomic assays as it marked the initial detection of significant antifungal activity. By selecting this specific time point, we aimed to detect the early molecular events associated with the observed antifungal effects of OPI.
So it is still 12 h in the text (now line228) not 9. So please decide from which hour the change was significant- from 9 or from 12?
-
Line 270: what are the proteins? Because in the table, only one protein alcohol dehydrogenase 1 catches the eye.
Response: Thank you for the correction. In line 270 the phrase “Once we observed the increase in the abundance of proteins involved in ethanol production, we performed the ethanol measurements in cells exposed to OPI and compared them to control cells” was changed to “Once we observed the increase in the abundance of alcohol dehydrogenase 1, we performed the ethanol measurements in cells exposed to OPI and compared them to control cells.
OK
Author Response
1. Fold changes below and above 2 seem insignificant in proteomic studies and the cut-off point 1.3 speaks of very small changes. More importantly, the lack of statistics with such small differences disqualifies these data as valid. It cannot be ruled out that if a statistical analysis were carried out, it would turn out that most of the differences are not significant.
Authors should necessarily provide statistical analysis. In addition, differences of the order of max. 2.3- fold raise the question of whether the Authors performed MS analyzes at the right time after OPI induction. OPI definitely affects the functioning of cells, hence the observed changes seem to be inadequately small.
Have the Authors deposited the original MS data?
It would be worth presenting a list of proteins in the supplement, the level of which did not change.
Response 1: Thank you for your feedback on the fold changes observed in our proteomic study. We acknowledge that in proteomic studies, fold changes below 2 may generally be considered less significant. However, it is worth noting that a cut-off point of 1.3 has been used in other research articles and can hold relevance in a broader context. Here are some examples of recent studies that have used a fold change of 1.3 as a cut-off point in proteomic analyses.
- Silva L.C. et al. Proteomic Response of Paracoccidioides brasiliensisExposed to the Antifungal 4-Methoxynaphthalene-N-acylhydrazone Reveals Alteration in Metabolism. J. fungi. doi: 10.3390/jof9010066.
- Marcos Abc Júnior M.A et al. Proteomic identification of metabolic changes in Paracoccidioides brasiliensis induced by a nitroheteroarylchalcone. Future Microbiol. doi: 10.2217/fmb-2022-0150.
- Lan J. et al. Quantitative Proteomic Analysis Uncovers the Mediation of Endoplasmic Reticulum Stress-Induced Autophagy in DHAV-1-Infected DEF Cells. 2019. Int J Mol Sci. doi: 3390/ijms20246160.
Of course, there are publications showing such small differences, but it still rather means that the experiment was poorly designed. In addition, 2 of the 3 papers listed are from the same team as the Authors. Such small changes indicate a small effect of the substance on the proteome, which is not in line with the other results. Also, the third paper cited by Lan shows a level of statistical significance - your paper does not. For me, these results are still not credible.
Response 2: We appreciate the discussion. As we stated previously but now with more detailed information, the protein identifications and quantitative packaging were generated using specific algorithms (Silva et al., 2005; Silva et al., 2006) and search was performed against a Paracoccidioides brasiliensis specific database. The ProteinLynx Global server v.2.5.2 (PLGS) with ExpressionE informatics v.2.5.2 was used to proper spectral processing, database searching conditions and quantitative comparisons. The identified proteins are organized by the expression algorithm into a statistically significant list corresponding to induced and reduced regulation ratios between infection and control groups. In the user’s guide, the ratio is showed as a probability of regulation. The software show the expression analysis statistics as the induced proteins with probability of upregulation 0.95 or more; the reduced proteins with probability of 0.05 or less. A value of 1.00 indicates that the cluster is definitely upregulated; a value of 0.00 indicates that the cluster is definitely downregulated. So the mathematical model used to calculate the ratios is a part of the Expression algorithm inside the PLGS from Waters Corporation (Geromanos et al., 2009). The protein/peptide scoring of the PLGS is a multistep process. Different mathematical models are used to match spectrum data to a protein in databank. The scores showed in the table are the sum of peptide and fragment scores for the data assigned to a protein. The data used in the manuscript considered only the protein identification with a confidence level of 95% and a false discovery rate of 4%. Those parameters for protein identification and expression are well established and accepted by the scientific community (Curty et al., 2014; Pizzatti et al., 2012).
Curty, N.; Kubitschek-Barreira, P. H.; Neves, G. W.; Gomes, D.; Pizzatti, L.; Abdelhay, E.; Souza, G. H.; Lopes-Bezerra, L. M. Discovering the infectome of human endothelial cells challenged with Aspergillus fumigatus applying a mass spectrometry label-free approach. J Proteomics 97: 126-140, 2014.
Geromanos, S. J.; Vissers, J. P.; Silva, J. C.; Dorschel, C. A.; LI, G. Z.; Gorenstein, M. V.; Bateman, R. H.; Langridge, J. I. The detection, correlation, and comparison of peptide precursor and product ions from data independent LC-MS with data dependant LC-MS/MS. Proteomics 9(6): 1683-1695, 2009.
Pizzatti, L., Panis, C., Lemos, G., Rocha, M., Cecchini, R., Souza, G.H., and Abdelhay, E. (2012). Label-free MSE proteomic analysis of chronic myeloid leukemia bone marrow plasma: disclosing new insights from therapy resistance. Proteomics 12, 2618-2631.
Silva, JC, Denny, R, Dorschel, CA, Gorenstein, M, Kass, IJ, Li, GZ, McKenna, T, Nold, MJ, Richardson, K, Young, P and Geromanos, S (2005). Quantitative proteomic analysis by accurate mass retention time pairs. Anal Chem 77(7): 2187-2200.
Silva, JC, Gorenstein, MV, Li, GZ, Vissers, JP and Geromanos, SJ (2006). Absolute quantification of proteins by LCMSE: a virtue of parallel MS acquisition. Mol Cell Proteomics 5(1): 144-156.
Response 1: Taking into consideration your remarks about the need to provide a statistical analysis, we would like to clarify that during the data processing of our mass spectrometry analyses, we used ProteinLynx Global Server version 2 software. This software generates a statistical analysis, which we employed as a protocol for the initial classification of induced or repressed proteins. In this protocol, we consider proteins as repressed when their values fall within the range of 0 to 0.5 and as induced when they are within the range of 0.95 to 1. Additionally, we also applied the classification by fold change to reinforce the analysis. We have added the statistical information in Supplementary Table 1 and 2.
A protocol saying that, for example, repressed proteins have fold values in the range of 0 to 0.5 is not a statistic! It’s just an estimation of fold change. In tables 1 and 2 from the supplement there is not a word about statistics. What was the p value? If the software perfomed statistical analysis, what was the number of proteins for which the fold change was not statistically significant? Normally there is a plenty of identified proteins in such analysis, but the significant change is only for some of them. The presented data is still unreliable without presented statistical analysis. If the Authors cannot say how was statistical analysis performed then they should use different tool to perform statistical analysis.
Response 2: All statistical analysis were performed by ExpressionE software v3.0 package (Waters, UK). The ion detection spectra counting, clustering and log-scale parametric normalisation procedures were performed into PLGS with ExpressionE license installed. The calculation of the log ratio and the confidence interval was based on a Gaussian distribution model, which allows for the possibility of an uncertain peptide assignment, an incorrect assignment of data to a cluster or an interference. The confidence interval of 95% was used.
- One of the conclusions of this work is that OPI induces changes in membrane structure. To make the work more complete, a proteomic analysis of membrane proteins would be useful to check whether their amount changes under the influence of OPI.
Response1 : We sincerely appreciate your valuable comment and suggestion regarding the inclusion of a proteomic analysis to investigate changes in membrane protein quantities under the influence of OPI in our work. However, a proteomic analysis is a complex task that requires proper planning, resources, and substantial time to be executed. Due to the scope of this article and other limitations, it is not feasible to conduct this specific analysis at the moment. Nonetheless, we will take this recommendation into consideration for future studies.
I still think that this experiment should be included if the Authors want to “sell” their work in good journal.
Response 2: We thank you for your consideration. As the focus of the present article, at first, was not to highlight the amount of membrane proteins under the influence of the compound, we did not perform this experiment, the procedure to extract Paracoccidioides membrane proteins is time consuming and we plan to investigate this possible alteration soon in a new approach.
- Ergosterol production- Figure 4b shows a greater than 50% increase in membrane ergosterol in cells treated with OPI for 48 hours. Only figure 1 shows that at 48 hours more than half of the cells treated with OPI are dead. Were the dead cells separated from the dead ones? If not, then the conclusion “However, after a prolonged period of exposure, Paracoccidioides may develop adaptations to counteract the effects of OPI on the membrane, such as increased ergosterol production which may contribute to the restoration of the membrane’s normal lipid composition” does not necessarily have to be accurate because it is not known whether obtained data refer to living or dead cells.
Response 1: Considering the limitation of the available data and the lack of distinction between living and dead cells after 48 hours of OPI treatment, we acknowledge that the original conclusion might be premature in stating that Paracoccidioides may develop adaptations to counteract the effects of OPI on the membrane, such as increased ergosterol production. Therefore, we have made the suggested modification to the sentence in line 304: "However, after a prolonged period of exposure, Paracoccidioides increased ergosterol production which may contribute to the restoration of the membrane's normal lipid composition."
It sounds a bit better, but still we don’t know if the higher ergosterol production may contribute to the restoration of the mebrane normal lipid composition, because it still seems that ergosterol level is corelated with the percentage of dead cells- so, higher ergosterol didn’t rescue the cells form death, hence probably yhe restoration was a failure.
Response 2: We agree with your assessment that the correlation between ergosterol levels and the percentage of dead cells raises questions about the effectiveness of elevated ergosterol in preventing cell death. We believe that this was not sufficient for membrane restoration, as the viability of P. brasiliensis continued to decrease over time Therefore, we have clarified this inthe manuscript. “We observed an increase in the ergosterol content in cells exposed to the antifungal after 48 h, which can be explained by compensatory mechanisms or alternative regulatory pathways that occur in response to the interruption of the sterol synthesis pathway. However, we believe that this was not sufficient for membrane restoration, as the viability of P. brasiliensis continued to decrease over time.”
Minor issues:
- Line 150: these 3 replicates are technical replicates after digestion or biological replicates? If after digestion, how many biological replicates were there or from how many mixed biological replicates were the proteins for MS isolated?
Response 1: We performed biological triplicates in our study. The fungus Paracoccidioides brasiliensis was incubated in the presence and absence of the compound on different days. After obtaining the triplicates, the samples were pooled and subsequently digested. The resulting pool was analyzed in triplicate in the mass spectrometry analyses.
This information should be incorporated into the text
Response 2: The information has been inserted into the manuscript.
- Line 211: what is the working principle of this reagent?
Response 1: The working principle of dichlorodihydrofluorescein 2,7-diacetate (DCFH-DA) dye is based on its ability to detect reactive oxygen species (ROS) within cells. DCFH-DA is a non-fluorescent compound that easily permeates cell membranes. Once inside the cell, it is deacetylated by intracellular esterases, converting it into the non-fluorescent dichlorodihydrofluorescein (DCFH). In the presence of ROS, such as hydrogen peroxide or hydroxyl radicals, DCFH undergoes oxidation, converting it into the highly fluorescent compound 2',7'-dichlorofluorescein (DCF). The intensity of the fluorescence is directly proportional to the amount of ROS present in the cell, allowing researchers to quantify the level of oxidative stress or assess ROS production in various experimental conditions.
I know that- the readers may not, so the summary of it should be pointed in the text .
Response 2: The information has been inserted into the manuscript.
- Line 222: Is the viability changed significantly from 12 or 9 hours? Because the statistically significant result is marked on the graph from the 9th hour of incubation with OPI.
Response 1: Thank you for your question. Indeed, we observed a statistically significant change in viability starting from the 9th hour of incubation with OPI. Therefore, we chose this time point for the proteomic assays as it marked the initial detection of significant antifungal activity. By selecting this specific time point, we aimed to detect the early molecular events associated with the observed antifungal effects of OPI.
So it is still 12 h in the text (now line228) not 9. So please decide from which hour the change was significant- from 9 or from 12?
Response 2: The information has been corrected into the manuscript. The viability was significantly affected after 9 h.